# A Medium- and Long-Term Orderly Charging Load Planning Method for Electric Vehicles in Residential Areas

**Zhaoxia Xiao** [1,*] **, Yi Zhou** [1] **, Jianing Cao** [2] **and Rui Xu** [1]

[1] School of Electrical Engineering, Tiangong University, Tianjin 300387, China; zhouyiburning@163.com (Y.Z.); xurui9803@163.com (R.X.)
[2] School of Control Science and Engineering, Tiangong University, Tianjin 300387, China; caojia_ning@163.com
* Correspondence: xiaozhaoxia@tiangong.edu.cn

**Abstract:** Due to the large number of electric vehicles (EVs) connected to the distribution network of residential areas (RAs), community charging has become a major constraint. The planning of the distribution network in RAs needs to consider the orderly charging load of EVs. In the current study, an orderly charging planning method for the charging posts and distribution network of RAs was proposed. First, a charging load forecasting model based on the travel characteristics, charging time, and ownership of EVs in RAs was established. Then, a hierarchical orderly charging optimization method, including a distribution network layer and EV access node layer, was devised. The upper layer optimizes the distribution network. The objective function is the minimum variance of the overall load in the RA and the constraint conditions satisfy the overall charging load demand and the capacity of the distributed network. The lower layer optimizes the EV access nodes. The objective function is the minimum variance of the node access load, and the constraint conditions are to meet the regional charging load demand and the optimal power balance demand transmitted from the upper layer to the lower layer. A nonlinear optimization algorithm is employed to solve these objective functions. An IEEE 33 node example was used to obtain the orderly charging power load curves for weekdays and weekends in RAs, and the simulation results prove the effectiveness of the proposed method.

**Keywords:** electric vehicles; residential areas; orderly charging; hierarchical optimization; nonlinear optimization algorithm

## 1. Introduction

Electric vehicles (EVs), as a means of replacing gasoline/diesel with electric energy, can help energy systems reduce their carbon emissions and achieve "carbon peak and carbon neutrality" [1]. Therefore, EVs have undergone rapid development in recent years. At present, EVs are mainly used for urban commuting in China. Further development of EVs is restricted by their lack of mileage and charging facilities. In particular, charging in residential areas (RAs) has become a bottleneck. Chinese urban residents mainly live in RAs, and private EVs are mainly charged in RAs. However, in the past, the distribution network planning of China's RAs did not consider the charging requirement of EVs. The large-scale connection of EVs to the distribution network of RAs will accelerate the demand for expansion and transformation of the distribution network, especially in old RAs. Nonetheless, charging facilities are gradually becoming more intelligent, and orderly charging has been implemented in the operation of the distribution network. Therefore, forecasting the charging load and analyzing the impact of orderly charging on the overall load level in the distribution network planning of RAs can assist in the planning of charging facilities and the required distribution network. This can effectively reduce the investment in construction or capacity expansion of the residential distribution network.

The forecasting of the EV charging load is the basis for the coordinated planning of charging facilities and distribution networks in RAs. It is mainly divided into short-term

load forecasting, and medium- and long-term load forecasting. In short-term forecasting of the charging load, a large amount of historical data of the charging point is collected, and the charging load characteristics of EVs are analyzed, to obtain the forecasted charging load power [2–5]. This is usually used for the optimal operation of the distribution network. In medium- and long-term forecasting of the charging load, a mathematical model is established based on the level of economic development, travel characteristics, and charging habits, and then used to predict the EV charging load power for the next few years [6–9]. This approach is mainly used for the planning and construction of the distribution network. The charging behavior of EVs is random in time and space, and the charging load power in different scenarios has different characteristics. The characteristics of the EV load include two main aspects: (1) the time distribution characteristics of the EV charging load; and (2) the space distribution characteristics of the EV charging load. Furthermore, EV ownership forecast is an important part of forecasting the charging load. Three models can be employed: the discrete model, the multi-agent model, and the diffusion model [10]. The discrete model constructs a utility function to describe a specific individual's preference for vehicle types, and further calculates the probability of consumers choosing each vehicle type. The multi-agent model takes individuals as the research object to model the response and mutual influence of individuals in different market environments. The diffusion model dynamically evolves the market share after the birth of a new product from a macro perspective. Diffusion models can be further divided into the Bass model, the Gompertz model, and the Logistic model. The Bass model is mainly affected by two factors, namely, external media spreading and internal oral spreading. For ownership forecasting of EVs in RAs, the internal oral spreading factors can include the purchase cost of EVs, mileage anxiety, the availability of charging facilities, and the visibility of electric vehicles. Therefore, the Bass model is used to predict the medium- and long-term EV ownership of RAs in this paper.

Orderly charging has been widely used in the optimal operation of distribution networks because it can reduce the network's peak-to-valley load [11]. However, in the current distribution network planning, the impact of orderly charging on the overall load level of the distribution network is rarely considered. For newly-built RAs, the EV load power is mainly used for distribution network planning with saturated load power, and the planning margin is too large, resulting in a waste of investment. For old RAs, a disorderly charging load is mainly used for distribution network planning, which leads to an increase in the peak load and difficulties in expanding the capacity of the residential distribution network. Therefore, by fully considering the impact of orderly charging loads in distribution network planning, an accurate and reasonable construction of the distribution network can be achieved. The orderly charging optimization strategy of EVs can be divided into two categories, that is, the direct optimization method and the indirect optimization method [12]. The direct optimization method directly controls the charging power and start time of EVs to meet the basic charging demand of EV users, and thus reduce power grid losses and peak-to-valley differences [13–17]. The indirect optimization method, that is, an electricity price mechanism, is used to guide users to actively adjust their charging behavior to maximize benefits of stakeholders [18–22].

In this paper, a charging load forecasting model based on the travel characteristics, charging time, and ownership of EVs in RAs is presented. First, Monte Carlo simulation is used to forecast future charging loads in RAs on weekdays and weekends. A hierarchical load optimization method including the distribution network layer and EV access node layer is presented. The objective function of the upper level minimizes the variance of the overall load in the RA, and the constraint conditions satisfy the overall charging load demand and the capacity of the distributed network. The objective function of the access node layer minimizes the load variance of the node access, and the constraint conditions are to meet the regional charging load demand and the optimal power balance demand transmitted from the upper layer to the lower layer. In addition, a nonlinear optimization algorithm is used to solve the orderly charging model, and an IEEE 33 node example is

used to obtain the orderly charging load curves for weekdays and weekends in RAs. Hence, the main goals of this paper as follows:

(1) To propose a forecasting mathematical model of the charging load for RAs;
(2) To develop a hierarchical load optimization method for analyzing the impact of orderly charging on the overall load level of the residential distribution network;
(3) To present a new typical analysis method for the daily load characteristics of urban RAs.

This paper is organized as follows. Section 2 introduces the structure of analyzing the impact of orderly charging on the overall load of RAs. Section 3 introduces the charging load forecasting of EVs in RAs. Section 4 describes the hierarchical optimization method, including load optimization of the distribution network layer and the EV access node layer. The case analysis of IEEE 33 node is presented in Section 5. Section 6 concludes this paper.

## 2. Architecture of Orderly Charging Load Planning in RAs

Figure 1 shows the architecture of mid-and long-term orderly charging load planning in RAs. The architecture mainly includes three parts: EV charging load forecasting in RAs, the hierarchical orderly charging strategy, and analysis of the impact of orderly charging on the overall load of distribution network in RAs. Firstly, the charging load forecast is obtained according to the ownership of EVs in RAs predicted by the Bass model and the charging probability curve of EVs in RAs. Then, based on the forecasted charging load, a hierarchical load optimization model for the distribution network of RAs considering the orderly charging of EVs is established. The minimum variance of the overall load of the distribution network is the upper-level optimization objective function and the minimum node load deviation is the lower-level optimization objective function. Therefore, the peak load growth level of the distribution network in RAs is the lowest considering the growth of the charging load. The nonlinear optimization algorithm is used to solve the hierarchical optimization model. The overall load optimization considering orderly charging in RAs during weekdays and weekends in the target year can be obtained.

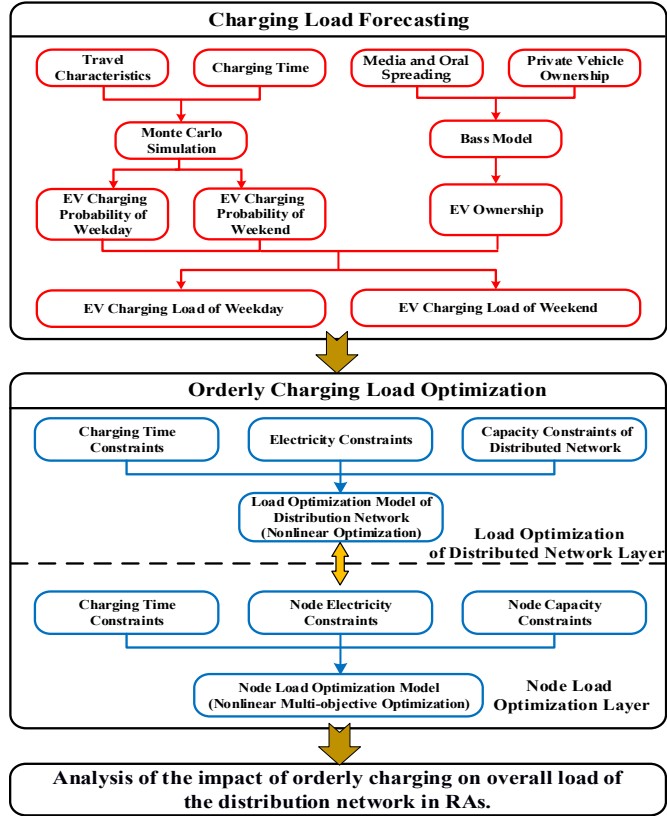

**Figure 1.** Architecture of mid-and long-term orderly charging load planning in RAs.

## 3. Forecasting EV Charging Load in RAs

The ownership of EVs and charging probability will affect the characteristics of EV charging load power. Therefore, a mathematical model to forecast the ownership of EVs in RAs is first established, and a charging probability model is also established considering the travel characteristics and charging time. Monte Carlo simulation is used to forecast the medium- and long-term charging load of EVs.

### 3.1. The Ownership of EVs

The ownership of EVs in RAs is affected by two main factors: external media spreading and internal oral spreading. A Bass model to forecast the ownership of EVs in RAs is established.

The expression of the Bass model is shown in Equation (1).

$$\frac{g(t)}{1 - G(t)} = p + qG(t) \tag{1}$$

where $G(t)$ is the proportion of total accumulated products in the largest market potential at time $t$; $g(t)$ is the proportion of new products at time $t$; $p$ is the innovation coefficient, which reflects the degree of influence of media spreading on the ownership of new products; $q$ is the imitation coefficient, which reflects the degree of influence of oral spreading on the ownership of new products. In this paper, the innovation coefficient $p$ of the Bass model of EVs is 0.03, and the imitation coefficient $q$ is 0.38 [23].

The private vehicle growth rate function is fitted by the private vehicle ownership data in a certain city [24]. It is shown in Equation (2).

$$f(x) = \alpha e^{-\left(\frac{t-\beta}{\gamma}\right)^2} \tag{2}$$

where $\alpha = 0.4652$, $\beta = 2003$, $\gamma = 12.51$.

### 3.2. Charging Probability Model

I.     Travel Characteristics

According to the family vehicle data from the department of transportation's survey, the characteristics of private vehicle travel on weekdays and weekends are obtained separately [25]. The probability density function at the work travel end time in the weekdays is shown in Equation (3).

$$f_{wd\text{-}w}(t) = \frac{1}{\sigma_{wd\text{-}w}\sqrt{2\pi}} e^{-\frac{(t-\mu_{wd\text{-}w})^2}{2\sigma_{wd\text{-}w}^2}} \tag{3}$$

where $\sigma_{wd\text{-}w} = 1.747$, $\mu_{wd\text{-}w} = 17.3$.

The probability density function at the end time of the weekday shopping and social travel is shown in Equation (4).

$$f_{wd\text{-}s}(t) = \frac{a_1}{\sigma_{wd\text{-}s1}\sqrt{2\pi}} e^{-\frac{(t-\mu_{wd\text{-}s1})^2}{2\sigma_{wd\text{-}s1}^2}} + \frac{b_1}{\sigma_{wd\text{-}s2}\sqrt{2\pi}} e^{-\frac{(t-\mu_{wd\text{-}s2})^2}{2\sigma_{wd\text{-}s2}^2}} \tag{4}$$

where $a_1 = 0.3841$, $\sigma_{wd\text{-}s1} = 2.32$, $\mu_{wd\text{-}s1} = 12$; $b_1 = 0.59$, $\sigma_{wd\text{-}s2} = 2.575$, $\mu_{wd\text{-}s2} = 18.2$.

The probability density function at the end time of the weekend shopping and social travel is shown in Equation (5).

$$f_{we\text{-}s} = \frac{a_2}{\sigma_{we\text{-}s1}\sqrt{2\pi}} e^{-\frac{(t-\mu_{we\text{-}s1})^2}{2\sigma_{we\text{-}s1}^2}} + \frac{b_2}{\sigma_{we\text{-}s2}\sqrt{2\pi}} e^{-\frac{(t-\mu_{we\text{-}s2})^2}{2\sigma_{we\text{-}s2}^2}} \tag{5}$$

where $a_2 = 0.302$, $\sigma_{we\text{-}s1} = 2$, $\mu_{we\text{-}s1} = 11.6$, $b_2 = 0.6395$, $\sigma_{we\text{-}s2} = 3.2$, $\mu_{we\text{-}s2} = 17$.

The probability density function of the mileage of a single trip of a vehicle is expressed as Equation (6).

$$L_s(l) = \frac{1}{l\sigma_s\sqrt{2\pi}} e^{-\frac{(\ln l - \mu_s)^2}{2\sigma_s^2}} \qquad (6)$$

where $l$ is the mileage of the vehicle in a single trip, in km; $\mu_s$ is the expectation value of $\ln l$; $\sigma_s$ is the standard deviation of $\ln l$ [2]. In this paper, the average travel distance of a single trip on weekdays is 11.4 km with a standard deviation of 4.88 km; the average mileage of a single trip on weekends is 13.2 km with a standard deviation of 5.23 km [24].

The initial state of charge (SOC) for EV charging can be expressed as Equation (7).

$$SOC_0 = (1 - \sum_{j=1}^{n} \frac{d_j}{D}) \times 100\% \qquad (7)$$

where $SOC_0$ represents the starting SOC of EV; $d_j$ represents the mileage of the $j$-th trip; $D$ represents the maximum mileage of the EV.

II.    Charging Time

The formula for calculating the charging time of an EV is shown in Equation (8).

$$t_c = \frac{(1 - SOC_0)E}{P_c} \qquad (8)$$

where $t_c$ is the charging time of the EV in h; $P_c$ is the charging power in kW; $E$ is the battery capacity of the EV in kW·h.

### 3.3. Forecasting of Charging Load

According to the established probability model of charging start time, charging start SOC, and charging duration, a large number of calculation samples are generated. The Monte Carlo simulation method is used to calculate the charging status of each calculation sample for a 24 h day, and to solve the charging probability curve of RAs during weekdays and weekends. Multiplying the prediction results of the ownership of EVs in RAs with the charging probability curve, the number of EVs in a charging state at each moment is calculated, and the number of EVs charged at each moment is multiplied by the charging power of EVs to obtain the EV charging load power of the RAs.

The steps of EV charging load prediction are as follows:

(1) The maximum market potential, innovation coefficient $p$, and imitation coefficient $q$ of EVs are selected, and the Bass model is used to iterate the medium- and long-term ownership of EVs in RAs;

(2) According to the established mathematical model of EV travel characteristics, a large number of samples are randomly selected;

(3) Using Equations (7) and (8), the charging start SOC and charging time of each random sample is calculated;

(4) Determine whether each random sample is in the charging state at each moment;

(5) According to the results of step (4), the daily charging probability curve of EVs in RAs is obtained;

(6) Cycle the above steps (2) to (5) 100 times to obtain the average value curve of the daily charging probability of EVs;

(7) Multiply the number of medium- and long-term EVs obtained in step (1) with the EV charging probability obtained in step (6), to obtain the number of charging EVs at each moment;

(8) The number of charging EVs at each moment in step (7) is multiplied by the charging power to obtain the EV charging load power in RAs.

## 4. Orderly Charging Strategy for EVs in RAs

A hierarchical load optimization method, including the distribution network layer in RAs and the lower layer of EV access nodes, is shown in Figure 2. The objective function of the upper level is to minimize the overall load variance of the RA, and the constraint conditions meet the overall charging load demand and the capacity of the distribution network. The objective function of each access node in the lower layer is to minimize the load variance of node access, and the constraint condition is to meet the charging load demand of the charging area, the node capacity requirement, and the optimal load power balance for each time period transmitted from the upper layer to the lower layer. If the lower-layer access node cannot meet the optimized load power balance required by the upper layer, it needs to feed back to the upper layer, requiring adjustment of the access node or adjustment of the optimized load power.

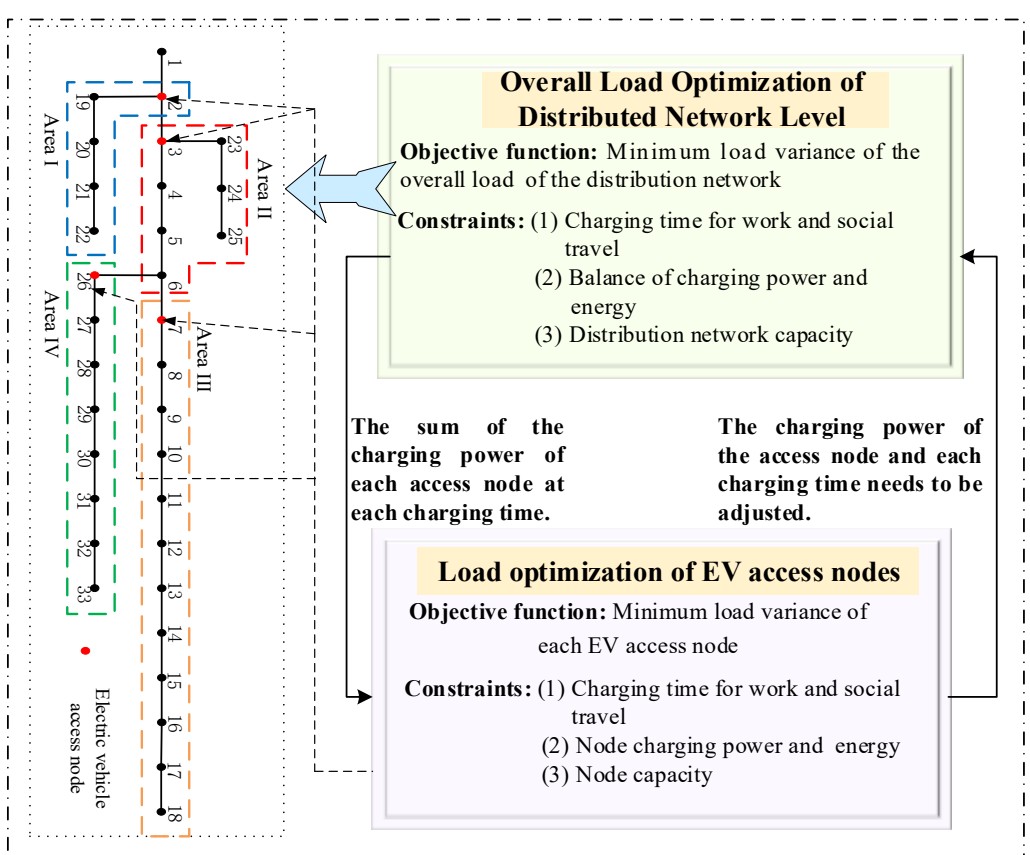

**Figure 2.** Hierarchical optimization model.

### 4.1. Orderly Charging Optimization of Distribution Network

I.   Objective Function

By shifting the charging load of EVs, it is possible to reduce the load peak of the distribution network in RAs.

The objective function of orderly charging of the overall charging load in the RA is shown in Equation (9).

$$\min F = \frac{1}{24}\sum_{i=1}^{24} \left(P_{EV}(i) + P_{norm}(i) - \overline{P}\right)^2 \tag{9}$$

where $P_{EV}(i)$ is the EV charging load power at the $i$-th hour; $P_{norm}(i)$ is the conventional daily load power of the RA at the $i$-th hour; $P$ is the average value of the overall load power of the RA.

II.  Constraints

(1)  Charging time constraints

The departure time for working travel is generally concentrated at 8:00 to 9:00, and the departure time for shopping–social travel is generally concentrated at 10:00. This article considers two charging time constraints for work trips and shopping–social trips.

The EV charging time constraint for workday travel is shown in Equation (10).

$$t_{ww\text{-}open} \leq t \leq t_{ww\text{-}end} \tag{10}$$

where $t_{ww\text{-}open}$ is the start time of orderly charging of EVs for work travel during workdays, and obeys the probability distribution at the end time of work travel during workdays, that is, Equation (3); $t_{ww\text{-}end}$ is the end time of orderly charging of EVs for work travels, and the value in this paper is 7:00.

The EV charging time constraint for shopping and social travel on weekdays is shown in Equation (11).

$$t_{ss1\text{-}open} \leq t \leq t_{ss1\text{-}end} \tag{11}$$

where $t_{ss1\text{-}open}$ is the start time of orderly charging of EVs for shopping–social travel on weekdays, and obeys the probability distribution of the end time of shopping-social travel on weekdays, that is, Equation (4); $t_{ss1\text{-}end}$ is the end time of orderly charging of EVs for shopping–social travel, and the value in this paper is 10:00.

The EV charging time constraint for shopping–social travel on weekends is shown in Equation (12).

$$t_{ss2\text{-}open} \leq t \leq t_{ss2\text{-}end} \tag{12}$$

where $t_{ss2\text{-}open}$ is the start time of orderly charging of EVs for shopping–social travel on weekends, and obeys the probability distribution of the end time of shopping–social travel on weekdays, that is, Equation (5); $t_{ss2\text{-}end}$ is the end time of orderly charging of EVs for shopping–social travel, and the value in this paper is 10:00.

(2)  Capacity constraints

The total load of the RA optimized through orderly charging should not exceed the capacity of the RA distribution network.

Therefore, the orderly charging capacity constraint is shown in Equation (13).

$$P_{EV}(i) + P_{norm}(i) \leq C_{\max} \tag{13}$$

where $P_{EV}(i)$ is the total charging load power of the EV at the *i*-th hour; $C_{max}$ is the maximum capacity of the residential distribution network.

(3)  Electricity constraints

The EV charging electricity is equal before and after the orderly charging optimization. Therefore, the orderly charging electricity constraint is shown in Equation (14).

$$\sum_{i=1}^{24} E_{EV}(i) = E_{sum} \tag{14}$$

where $E_{EV}(i)$ is the EV charging electricity at the *i*-th hour; $E_{sum}$ is the total charging electricity.

*4.2. Node Orderly Charging Optimization*

The optimization results obtained by the orderly charging optimization of the distribution network layer are not the best results for the charging nodes. Therefore, the overall load of each charging node needs to be optimized again.

I.  Objective Function

The objective function of each charging node is shown in Equation (15).

$$
\begin{cases}
\min f = [f_1, f_2, \ldots, f_j, \ldots, f_n] \\
f_j = \dfrac{1}{24} \displaystyle\sum_{i=1}^{24} \left( p_{EV\text{-}j}(i) + p_{norm\text{-}j}(i) - \overline{p}_j \right)^2
\end{cases}
\tag{15}
$$

where $f_j$ is the overall load variance of the node $j$; $p_{EV\text{-}j}$ is the EV charging load of the node $j$ at the $i$-th hour; $p_{norm\text{-}j}$ is the conventional daily load of the node $j$ at the $i$-th hour; $\overline{p}_j$ is the load mean value of the node $j$ in the RA.

II.  Constraints

(1)  Charging time constraints
The orderly charging optimization load of each node should meet the same charging time constraints.
(2)  Electricity constraints
The EV charging electricity in each charging node is equal before and after the orderly charging optimization. Therefore, the orderly charging electricity constraint of the charging node is shown in Equation (16).

$$
\begin{cases}
\displaystyle\sum_{i=1}^{24} E_{EV\text{-}j}(i) = E_{sum\text{-}j} \\
\displaystyle\sum_{j=1}^{n} E_{sum\text{-}j} = E_{sum}
\end{cases}
\tag{16}
$$

where $E_{EV\text{-}j}(i)$ is the EV charging electricity of the node $j$ at the $i$-th hour; $E_{sum\text{-}j}$ is the sum of the charging electricity of the node $j$.

(3)  Node capacity constraints
The node load should not exceed the node capacity of the distribution network. Therefore, the capacity constraint of charging node is shown in Equation (17).

$$
p_{EV\text{-}j}(i) + p_{norm\text{-}j}(i) \leq C_{j\text{-max}}
\tag{17}
$$

where $C_{j\text{-}max}$ is the maximum capacity of the node $j$.

(4)  Charging power constraint
The superimposed orderly charging optimization curve of each charging node should be the same as the orderly charging optimized power of the distribution network.
Therefore, the charging power constraint of each charging node is shown in Equation (18).

$$
\sum_{j=1}^{n} p_{EV\text{-}j}(i) = P_{EV}(i)
\tag{18}
$$

*4.3. Nonlinear Optimization*

The orderly charging optimization models of EVs are typical nonlinear algebraic equations. The nonlinear optimization method is to solve the extreme value of an n-ary real function under the constraints of a set of inequalities or equalities. The results of EV load forecasting in RAs are obtained first, then the forecasted charging load is the load balance constraint of the orderly charging optimization model. The nonlinear optimization method is used to solve the orderly charging load power of the distribution network and each charging node.

## 5. Case Analysis

*5.1. Simulation Parameters*

An IEEE-33 node power distribution system is used as a typical case to verify the orderly charging planning strategy of EVs in RAs. The capacity reference value of IEEE-33 node power distribution network is set to 5 MVA, the reference voltage is set to 10 kV,

and its active and reactive load power load parameters and line impedances are shown in Appendix A (Tables A1 and A2). Node 1 is the power supply node and its node voltage has a per unit value of 1.05. The other nodes are all PQ nodes, and the allowed per unit value of the node voltage is 0.93–1.05.

By analyzing the daily load data of a RA in a certain province of China, the daily load power curve of typical weekdays and weekends in summer was obtained, as shown in Figure 3. This paper assumes that the load characteristics of each node in the IEEE-33 node distribution network meet the typical daily load curve. Multiplying the typical daily load factor with the load power data of the IEEE-33 node system, we can obtain the load power of each node at each time of the day.

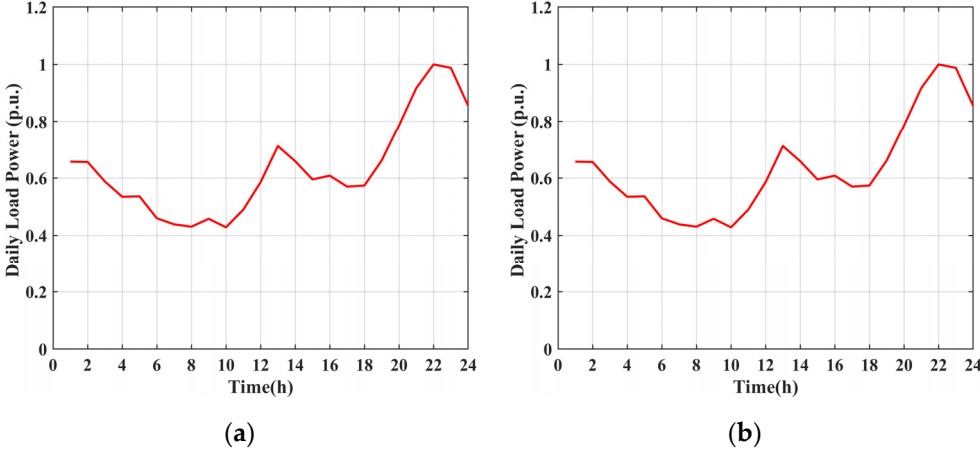

**Figure 3.** Typical daily load power per unit: (**a**) weekday (**b**) weekend.

The parameters of a typical RA are shown in Table 1. The parameters of "BYD Yuan EV" are used in this paper. The EV has a battery capacity of 40.62 kW·h and a driven distance of 305 km. On weekdays, the proportion of work travel is set to 80%, and the proportion of shopping–social travel is set to 20%. On weekends, the proportion of shopping–social travel is set to 100%. Furthermore, the travel probability of private vehicles is set to 77% on weekdays and 70% on weekends [26].

**Table 1.** Relevant parameters of the RA.

| Number of Households | Charging Power of a Single Charging Post | Number of Residents | Capacity of Distributed Network |
|---|---|---|---|
| 2000 | 7 kW | 6000 | 5000 kVA |

According to data collected by a provincial statistics bureau in China, it can be seen that the ownership of private vehicles per 1000 people in the region in 2019 was 338, and EVs accounted for about 2.05%. Therefore, in 2019, the ownership of private vehicles in this RA was 2028, and the ownership of EVs was 41. Using Equation (2) and the ownership of private vehicles in 2019, the saturation value of private vehicle ownership in the RA can be calculated as 2762. In this paper, the innovation coefficient $p$ of the Bass model of EVs was selected as 0.03, and the imitation coefficient $q$ was selected as 0.38. The maximum market potential of EVs is 70% of the saturation value of private vehicle ownership in RAs. Table 2 shows the forecast results of the ownership of EVs in RAs.

**Table 2.** Forecast results of the ownership of EVs in RAs.

| Years | 2021 | 2022 | 2023 | 2025 |
|---|---|---|---|---|
| EV ownership | 168 | 260 | 373 | 654 |
| Private vehicles | 2303 | 2409 | 2495 | 2618 |

### 5.2. Simulation Result

The Bass model and Monte Carlo simulation are used to predict the medium- and long-term charging load of EVs in RAs, and the objective function is to minimize the overall load variance of the RA distribution network, and the orderly charging load of EVs is calculated using a nonlinear optimization algorithm. The conventional load of RAs is predicted by the equal growth rate method, and the growth rate is set to 1%. By superimposing the orderly charging load curve with the conventional load curve, the overall load optimization curve of the RA distribution network in 2021, 2022, 2023, and the target year 2025 can be derived, as shown in Figure 4.

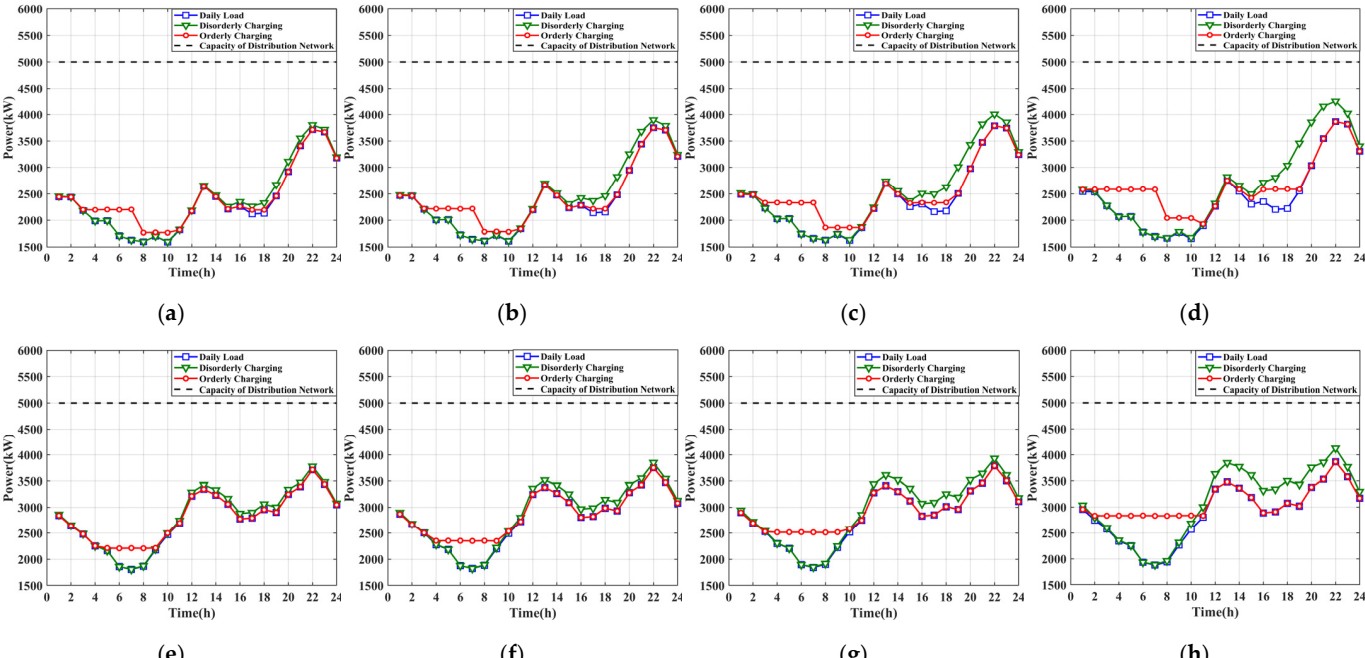

**Figure 4.** Orderly charging planning results: (**a**) 2021 (weekday), (**b**) 2022 (weekday), (**c**) 2023 (weekday), (**d**) 2025 (weekday), (**e**) 2021 (weekend), (**f**) 2022 (weekend), (**g**) 2023 (weekend), (**h**) 2025 (weekend).

From Figure 4a,e, it can be seen that there are fewer EVs in RAs, and disorderly charging has less impact on the distribution network of RAs. From Figure 4a–h, it can be seen that, with the continuous increase in the ownership of EVs in RAs, the impact of disorderly charging on the distribution network of RAs will gradually increase, and the peak load power of EVs will be added to the daily load power. By comparison, after orderly charging, the peak charging load can be moved to the valley period of the daily load power. By the target year 2025, when the charging load electricity accounts for about 10% of the daily load, it will not increase the peak load of the distribution network.

The peak-to-valley difference comparison between disorderly charging and orderly charging is shown in Table 3. With the increase in the charging load and conventional daily load, the peak-to-valley difference in disorderly charging will increase, accelerating the upgrading of the residential distribution network. The peak-to-valley difference in orderly charging is significantly lower than that in disorderly charging. Therefore, orderly charging can be used to reduce the peak load power of the distribution network in RAs and avoid possible heavy overload problems in the network.

**Table 3.** Comparison of load peak-to-valley difference between orderly and disorderly charging.

| Years | Peak-to-Valley Difference/kW | | | |
|---|---|---|---|---|
| | Disorderly Charging (Weekday) | Orderly Charging (Weekday) | Disorderly Charging (Weekend) | Orderly Charging (Weekend) |
| 2021 | 2208.1 | 1949.1 | 1971.8 | 1506.7 |
| 2022 | 2285.3 | 1970.3 | 2025.9 | 1403.8 |
| 2023 | 2375.5 | 1928.8 | 2080.2 | 1277.6 |
| 2025 | 2579.9 | 1937.3 | 2238.3 | 1035.6 |

Taking the overall load optimization data of the RA distribution network in 2025 as an example, the impact of EV charging load power on the RA distribution network was analyzed. The IEEE-33 node power distribution system is divided into four areas, among which nodes 2, 3, 7, and 26 are charging nodes for EVs, as shown in Figure 5. The charging load of each area is distributed in proportion to the daily load of this area.

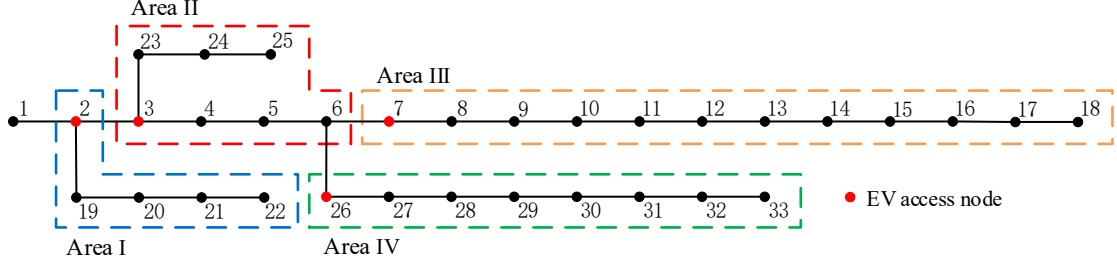

**Figure 5.** Area division of IEEE-33 node distribution network.

On the basis of satisfying the overall load optimization of the residential distribution network, the node loads of nodes 2, 3, 7, and 26 are optimized again with the minimum node load variance as the objective function. The optimization results are shown in Figure 6. It can be seen from Figure 6 that after the implementation of the orderly charging strategy, the peak load of the EV access node will shift. The maximum peak of the overall load of the charging node after orderly charging on weekdays appears at 7:00, and the maximum peak of the overall load on weekends appears at 7:00. From Figure 6e–h, it can be seen that the orderly charging peak power may be bigger than the peak power of disorderly charging. This is because the charging load power will be shifted after the implementation of the orderly charging strategy and the total charging load of each EV access node is required to meet the optimal charging load power transmitted from the upper layer to the lower layer. If the peak load power of each EV access node is over the node capacity, the number of EV access nodes will be increased. The orderly charging peak load of each access node can provide a data basis for the future planning of charging facilities and the transformation of distribution transformers in the station area in RAs.

Table 4 shows the comparison between the charging electricity of each access node and the overall load electricity. It can be seen from Table 4 that the charging electricity of each access node accounts for more than 20% of the total electricity of the node. Therefore, after the implementation of orderly charging, the overall load peak value of the EV access node may be higher than the load peak value of disorderly charging. From Figure 6, the overall load peak after orderly charging on weekends will be higher than that of disorderly charging. The reason for this is that the disorderly charging load power of EVs on weekends is more evenly distributed in time. After orderly charging, the charging load is concentrated in 1:00–10:00.

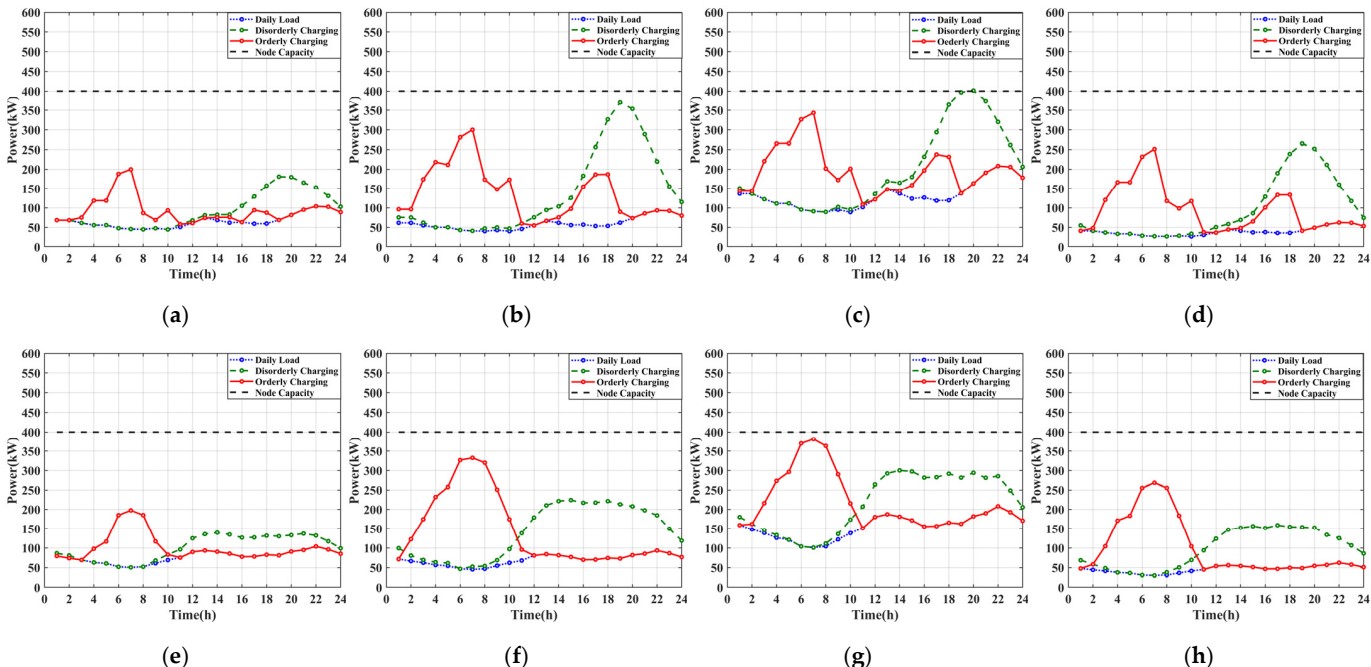

**Figure 6.** EV access node load power: (**a**) Node 2 (weekday), (**b**) Node 3 (weekday), (**c**) Node 7 (weekday), (**d**) Node 26 (weekday), (**e**) Node 2 (weekend), (**f**) Node 3 (weekend), (**g**) Node 7 (weekend), (**h**) Node 26 (weekend).

**Table 4.** Comparison of charging power of access nodes.

| Typical Day | Node 2 Overall Electricity (kWh) | Node 2 Charging Electricity (kWh) | Node 3 Overall Electricity (kWh) | Node 3 Charging Electricity (kWh) | Node 7 Overall Electricity (kWh) | Node 7 Charging Electricity (kWh) | Node 26 Overall Electricity (kWh) | Node 26 Charging Electricity (kWh) |
|---|---|---|---|---|---|---|---|---|
| Weekday | 2218 | 637 | 3271 | 1848 | 4716 | 1554 | 2285 | 1337 |
| Weekend | 2433 | 574 | 3395 | 1722 | 5181 | 1463 | 2368 | 1253 |

The voltage of the EV access node by orderly and disorderly charging is shown in Figure 7. It can be seen from Figure 7 that the node voltage drops after the node is connected to the EV load. After orderly charging optimization, the voltage offset of the EV access node is reduced overall, and the power quality is improved.

The voltages of the 18th, 22th, 25th, and 33rd nodes in the IEEE-33 distribution network are shown in Figure 8. It can be seen from Figure 8 that the farther the node from the power supply node, the greater the degree of node voltage deviation. When the 2nd, 3rd, 7th, and 26th nodes are connected to the EV load, the voltage drop will be more serious. If orderly charging is implemented, the voltage quality will be improved. In the IEEE-33 node system, the maximum voltage deviation node is the 18th node. Therefore, when considering the planning of charging facilities in RAs, we should avoid connecting the charging facilities at the end node, and the access point should be as close to the power supply node as possible.

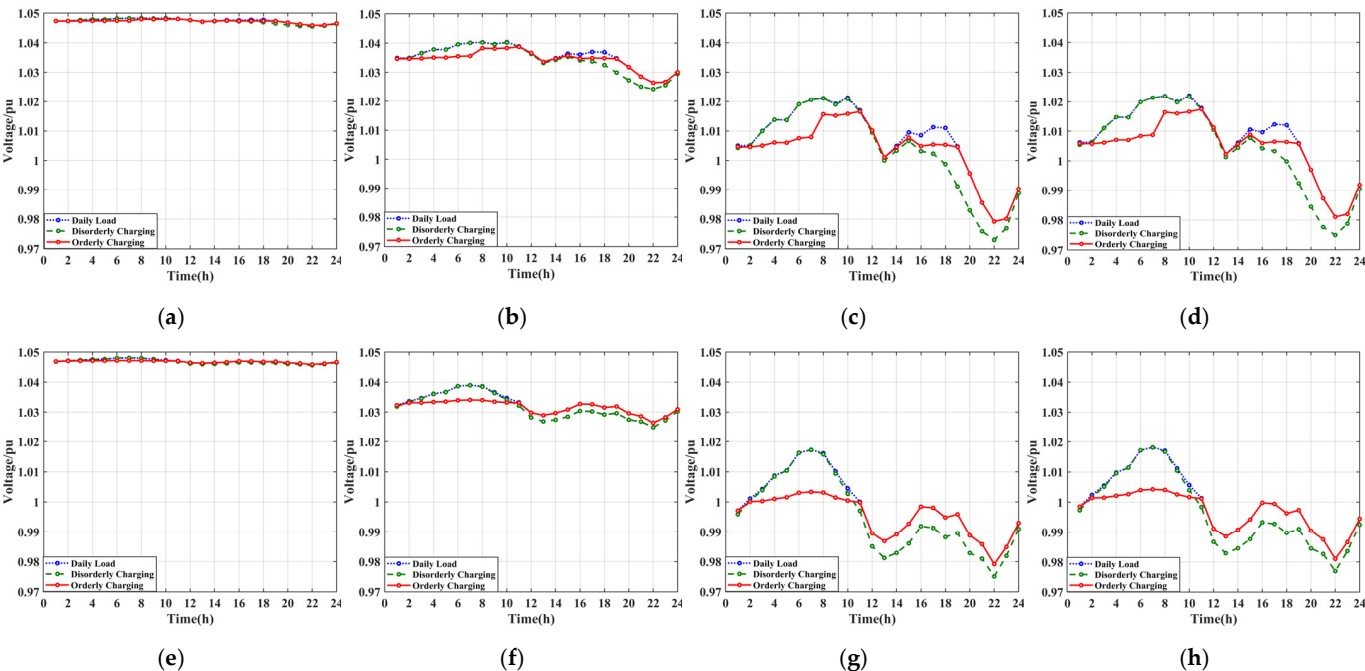

**Figure 7.** The voltage of the EV access nodes. (**a**) Node 2 (weekday), (**b**) Node 3 (weekday), (**c**) Node 7 (weekday), (**d**) Node 26 (weekday), (**e**) Node 2 (weekend), (**f**) Node 3 (weekend), (**g**) Node 7 (weekend), (**h**) Node 26 (weekend).

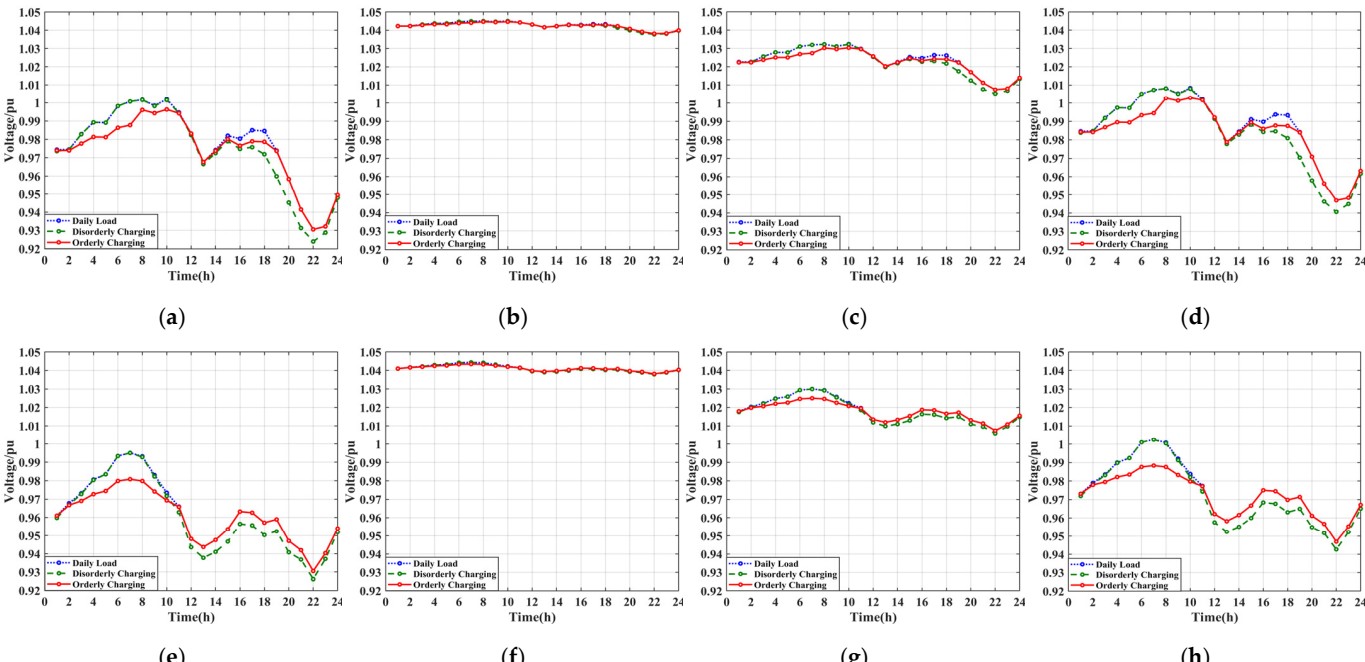

**Figure 8.** Voltage of nodes 18, 22, 25, and 33: (**a**) Node 18 (weekday), (**b**) Node 22 (weekday), (**c**) Node 25 (weekday), (**d**) Node 33 (weekday), (**e**) Node 18 (weekend), (**f**) Node 22 (weekend), (**g**) Node 25 (weekend), (**h**) Node 33 (weekend).

The voltages of the IEEE-33 distribution network are shown in Figure 9. The IEEE-33 node voltages at 19:00 and 22:00 on weekdays are shown in Figure 9a. The time of 19:00 is when the load difference between orderly charging and disorderly charging is the largest on a weekday; that is, the voltage characteristic curve at 19:00 can describe the maximum improvement of orderly charging to the voltage quality of the distribution network. The time of 22:00 is when the IEEE-33 node system load is the highest; that is, the voltage characteristic curve at 22:00 can describe the node voltage deviation in the most serious

situation. In the above two cases, orderly charging can improve the voltage quality. The same analysis can be undertaken in Figure 9b.

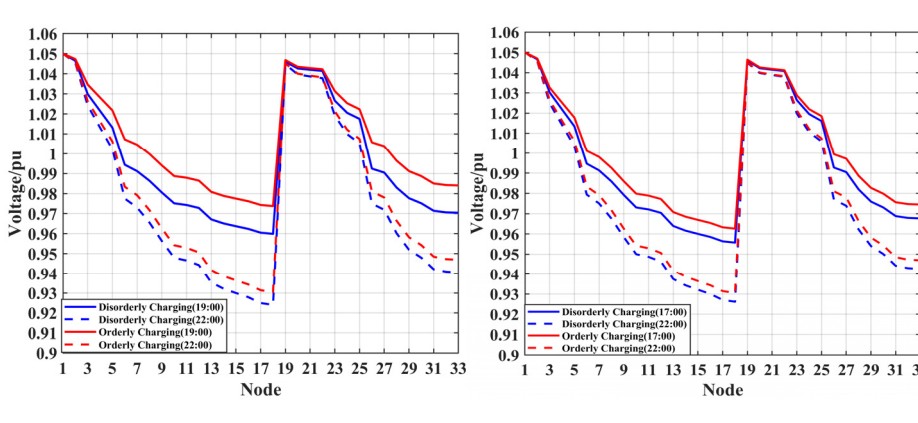

(**a**)                                          (**b**)

**Figure 9.** Voltages of the IEEE-33 distribution network: (**a**) weekday, (**b**) weekend.

Table 5 shows the comparison of the network loss between orderly charging and disorderly charging. By 2025, when the EV load electricity on weekdays and weekends accounts for 8.39% and 6.77%, respectively, of the overall residents' load electricity, if disorderly charging is used, the network loss will account for 4.15% and 4.72%; if orderly charging is used, the network loss will account for 3.97% and 4.57%. Therefore, the use of orderly charging can reduce the network loss. In the future, with the increasing proportion of EV load electricity, distribution network planning considering orderly charging can effectively save operation costs.

**Table 5.** Comparison of the network loss between orderly charging and disorderly charging.

| Typical Day | Daily Electricity (kWh) | EVs Charging Electricity (kWh) | Network Loss of Disorderly Charging (kWh) | Network Loss of Orderly Charging (kWh) |
|---|---|---|---|---|
| Weekday | 58,725.7 | 5376 | 2658 | 2545 |
| Weekend | 69,066.8 | 5012 | 3499 | 3388 |

## 6. Conclusions

An orderly charging method for planning the charging facilities and distribution network of RAs is proposed. A charging load forecasting model based on the travel characteristics, charging time, and ownership of EVs in RAs is first established. Then, a hierarchical optimization method including load optimization of the distribution network layer and each access node of EVs is presented. The nonlinear optimization algorithm is employed to solve these objective functions, and an IEEE 33 node case is used to obtain the orderly charging load power curves for weekdays and weekends in RAs. Thus, the following functionalities are achieved:

(1) For the optimization of the distribution network layer, when using orderly charging, the overall peak–valley difference and peak load of the RA does not exceed the daily peak load until the target year.
(2) For the optimization of each access node, the charging load power will be shifted after the implementation of the orderly charging strategy. The total peak load of the EV access node may be bigger than the peak power of disorderly charging depending on the number of EV access nodes and the shifted charging load power. The orderly charging load of each EV access node can provide a data basis for the future planning of charging facilities in RAs.

(3) Implementation of orderly charging can improve the voltage quality of the distribution network in RAs and can reduce the network loss. With the increasing proportion of EV charging electricity, distribution network planning considering orderly charging can effectively save investment and operation costs.

**Author Contributions:** Conceptualization, Z.X.; methodology, Z.X. and Y.Z.; software, Y.Z.; validation, Z.X. and Y.Z.; data curation, Y.Z.; writing—original draft preparation, Z.X. and Y.Z.; writing—review and editing, Z.X., Y.Z., J.C. and R.X. All authors have read and agreed to the published version of the manuscript.

**Funding:** This research was funded by National Natural Science Foundation of China, grant number 51977149. And this work was supported by Hebei University of Technology State Key Laboratory of Reliability and Intelligence of Electrical Equipment, grant number EERI_KF2020003.

**Institutional Review Board Statement:** Not applicable.

**Informed Consent Statement:** Not applicable.

**Data Availability Statement:** The dataset used in this article can be obtained from the corresponding author under reasonable request.

**Conflicts of Interest:** The authors declare no conflict of interest.

### Nomenclature

| Parameter | Description |
| --- | --- |
| $g$ | Ratio of newly added products to the largest market potential [-] |
| $G$ | Ratio of the total cumulative products to the largest market potential [-] |
| $p$ | Innovation coefficient [-] |
| $q$ | Imitation coefficient [-] |
| $h$ | Private vehicle ownership growth rate [-] |
| $f_{wd\text{-}w}$ | Probability density function at the work travel end time in the weekdays [-] |
| $f_{wd\text{-}s}$ | Probability density function at the end time of the weekday shopping and social travel [-] |
| $f_{we\text{-}s}$ | Probability density function at the end time of the weekend shopping and social travel [-] |
| $L$ | Probability density function of a single mileage [-] |
| $l$ | Mileage of the vehicle in a single travel [km] |
| $\mu_s$ | Expectation of $\ln l$ [km] |
| $\sigma_s$ | Standard deviation of $\ln l$ [km] |
| $SOC_0$ | Starting SOC of the electric vehicle charging [%] |
| $d_j$ | Mileage of the $j$-th travel [km] |
| $D$ | Maximum mileage of the electric vehicle [km] |
| $t_c$ | Charging time of the electric vehicle [hour] |
| $E$ | Battery capacity of the electric vehicle [kW·h] |
| $P_c$ | Charging power [kW] |
| $F$ | Objective function of the upper level [-] |
| $P_{EV}$ | Charging load [kW] |
| $P_{norm}$ | Conventional electricity load [kW] |
| $P$ | Average value of the overall load [kW] |
| $t_{ww\text{-}open}$ | Start time of orderly charging of electric vehicles for work travels on workdays [hour] |
| $t_{ww\text{-}end}$ | End time of orderly charging of electric vehicles for work travels on workdays [hour] |
| $t_{ss1\text{-}open}$ | Start time of orderly charging of electric vehicles for shopping and social travel on weekdays [hour] |
| $t_{ss1\text{-}end}$ | End time of orderly charging of electric vehicles for shopping and social travel on weekdays [hour] |
| $t_{ss2\text{-}open}$ | Start time of orderly charging of electric vehicles for shopping and social travel on weekends [hour] |

| | |
|---|---|
| $t_{ss2\text{-}end}$ | End time of orderly charging of electric vehicles for shopping and social travel on weekends [hour] |
| $C_{max}$ | Maximum capacity of the residential distribution network [kVA] |
| $E_{EV}$ | Electric vehicle charging electricity in each hour [kW·h] |
| $E_{sum}$ | Total charging electricity [kW·h] |
| $f_j$ | Load variance of the node $j$ [-] |
| $p_{EV\text{-}j}$ | Electric vehicle charging load of the node $j$ [kW] |
| $p_{norm\text{-}j}$ | Conventional daily load of the node $j$ [kW] |
| $P_j$ | Average value load of the node $j$ [kW] |
| $E_{EV\text{-}j}$ | Electric vehicle charging electricity of the node $j$ in each hour [kW·h] |
| $E_{sum\text{-}j}$ | Charging electricity of the node $j$ [kW·h] |
| $C_{j\text{-}max}$ | Maximum capacity of the node $j$ [kVA] |

## Appendix A

**Table A1.** Load parameters of IEEE-33 node power distribution system.

| Node Number | Node Injected Active Power (kW) | Node Injected Reactive Power (kVar) | Node Capacity(kVA) |
|:---:|:---:|:---:|:---:|
| 1 | - | - | - |
| 2 | 100 | 30 | 400 |
| 3 | 90 | 25 | 400 |
| 4 | 120 | 35 | 400 |
| 5 | 60 | 15 | 400 |
| 6 | 60 | 15 | 400 |
| 7 | 200 | 60 | 400 |
| 8 | 200 | 60 | 400 |
| 9 | 60 | 15 | 400 |
| 10 | 60 | 15 | 400 |
| 11 | 45 | 10 | 400 |
| 12 | 60 | 15 | 400 |
| 13 | 60 | 15 | 400 |
| 14 | 120 | 35 | 400 |
| 15 | 60 | 10 | 400 |
| 16 | 60 | 15 | 400 |
| 17 | 60 | 15 | 400 |
| 18 | 90 | 25 | 400 |
| 19 | 90 | 25 | 400 |
| 20 | 90 | 25 | 400 |
| 21 | 90 | 25 | 400 |
| 22 | 90 | 25 | 400 |
| 23 | 90 | 25 | 400 |
| 24 | 420 | 100 | 630 |
| 25 | 420 | 100 | 630 |
| 26 | 60 | 15 | 400 |
| 27 | 60 | 15 | 400 |
| 28 | 60 | 10 | 400 |
| 29 | 120 | 35 | 400 |
| 30 | 200 | 60 | 400 |
| 31 | 150 | 45 | 400 |
| 32 | 210 | 60 | 400 |
| 33 | 60 | 15 | 400 |

**Table A2.** Line parameters of IEEE-33 node power distribution system.

| Starting Node | End Node | Resistance ($\Omega$) | Reactance ($\Omega$) |
|---|---|---|---|
| 1 | 2 | 0.0922 | 0.047 |
| 2 | 3 | 0.493 | 0.2511 |
| 3 | 4 | 0.366 | 0.1864 |
| 4 | 5 | 0.3811 | 0.1941 |
| 5 | 6 | 0.819 | 0.707 |
| 6 | 7 | 0.1872 | 0.6188 |
| 7 | 8 | 0.7114 | 0.2351 |
| 8 | 9 | 1.03 | 0.74 |
| 9 | 10 | 1.044 | 0.74 |
| 10 | 11 | 0.1966 | 0.065 |
| 11 | 12 | 0.3744 | 0.1238 |
| 12 | 13 | 1.468 | 1.155 |
| 13 | 14 | 0.5416 | 0.7129 |
| 14 | 15 | 0.591 | 0.526 |
| 15 | 16 | 0.7463 | 0.545 |
| 16 | 17 | 1.289 | 1.721 |
| 17 | 18 | 0.732 | 0.574 |
| 2 | 19 | 0.164 | 0.1565 |
| 19 | 20 | 1.5042 | 1.3554 |
| 20 | 21 | 0.4095 | 0.4784 |
| 21 | 22 | 0.7089 | 0.9373 |
| 3 | 23 | 0.4512 | 0.3083 |
| 23 | 24 | 0.898 | 0.7091 |
| 24 | 25 | 0.896 | 0.7011 |
| 6 | 26 | 0.203 | 0.1034 |
| 26 | 27 | 0.2842 | 0.1447 |
| 27 | 28 | 1.059 | 0.9337 |
| 28 | 29 | 0.8042 | 0.7006 |
| 29 | 30 | 0.5075 | 0.2585 |
| 30 | 31 | 0.9744 | 0.963 |
| 31 | 32 | 0.3105 | 0.3619 |
| 32 | 33 | 0.341 | 0.5302 |

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
