# Peer review of "A Medium- and Long-Term Orderly Charging Load Planning Method for Electric Vehicles in Residential Areas"

_wevj, doi:10.3390/wevj12040216_

Round 1
Reviewer 1 Report
The authors propose a model for forecasting EVs charging demand in residential areas and uses this to develop a technique for orderly charging of EVs to reduce the impact of charging on the electricity grid.
Overall, the subject is interesting and work is technically sound, but needs some careful attention, as follows:
The authors seem to consider only two factors that affect ownership of EVs: “media propaganda and oral propaganda”. Surly there are other equally (if not more) important factors, e.g. cost, range anxiety, availability of charging infrastructure, etc. The authors need to consider these, or at least analysis and explain the impact they could have on the results obtained.
The use of English is not concise and there are many grammatical errors and instances of badly worded/constructed sentences. e.g., e.g. see the abstract. The manuscript needs a careful and professional proofread.
Need to clearly define all expressions and abbreviations the first time they are used; e.g. what is “Oral and Media” in Fig. 1? What is BASS in section 2?
Ras in lower box of Fig. 1 should be RAs.
Axes of graphs, e.g. Fig.3 should be appropriately labeled. Also, the demand profiles for the weekday and weekend in (a) and (b) seem a bit unusual. Shouldn’t these be the other way round, this is: (a) weekend and (b) weekday?
I have not come across the term “charging pile”. In the literature this is usually referred to as charging post, point or station.
What does “Combining (2)” on page 9 means?
Please check conclusions 2, Line 2: I think “greater” is wrong.
There are significant number of publications on the impact of EV charging on the grid. Authors are advised to enhance the literature review, to clearly demonstrate the added value and new knowledge that this paper presents. Just to give a few examples:
- Das R., et al, “Real-time multi-objective optimisation for electric vehicle charging management”, Elsevier journal of Cleaner Production, Vol. 292, 10 April 2021 126066, pp 1-19.
- Lacey G., et al, “Smart EV Charging Schedules: Supporting the Grid and Protecting Battery Life", IET Electrical Systems in Transportation, Vol. 7, Issue 1, March 2017, pp 84-91.
- Jiang T., et al, “Development of a Decentralized Smart Charge Controller for Electric Vehicles”, Elsevier International Journal of Electrical Power & Energy Systems, Volume 61, October 2014, pp. 355–370.
- Putrus G. A., et al, “Impacts of Electric Vehicles on Power Distribution Networks” IEEE Vehicle Power and Propulsion Conference, Michigan, Sept. 2009, pp 827-831.
Please note that these are only examples, and you may include what you believe is relevant. The important main point is to better discuss the state-of-the-art and to put the interpretation of the results into the right context.
Author Response
Manuscript ID: wevj-1400497
Title: A Medium and Long Term Orderly Charging Load Planning Method for Electric Vehicles in Residential Areas
Journal: World Electric Vehicle Journal
Mr. Nopparat Komrapit, Assistant Editor, World Electric Vehicle Journal
First of all, we would like to thank you, the assistant editor and the reviewers for your insightful and helpful comments. We have carefully read the comments from you and the reviewers, and we have revised our paper according to the comments and suggestions. The changes in the manuscript are marked up using the “Track Changes”.
The authors.
Reviewer 1:
The authors propose a model for forecasting EVs charging demand in residential areas and uses this to develop a technique for orderly charging of EVs to reduce the impact of charging on the electricity grid.
Overall, the subject is interesting and work is technically sound, but needs some careful attention, as follows:
Comments:
Comment 1: The authors seem to consider only two factors that affect ownership of EVs: “media propaganda and oral propaganda”. Surly there are other equally (if not more) important factors, e.g. cost, range anxiety, availability of charging infrastructure, etc. The authors need to consider these, or at least analysis and explain the impact they could have on the results obtained.
Response 1: The use of fuel vehicles is restricted in China due to environmental issues. Electric vehicles are mainly used for work and life travel of urban residents, and are rarely used for long-distance travel between cities in China. According to the actual demand of Chinese urban residents for electric vehicles, electric vehicles can be regarded as durable goods such as TVs and refrigerators. The diffusion process based on the inventory of durable goods is mainly affected by two factors: external media spreading and internal oral spreading. Therefore, these two main factors is considered for forecasting the ownership of electric vehicles in residential areas. At present, cost, mileage anxiety, and the availability of charging infrastructure are not the main factors affecting the ownership of electric vehicles in urban residential areas. With the development of electric vehicles in the future, we will develop the ownership forecast model of electric vehicle.
Comment 2: The use of English is not concise and there are many grammatical errors and instances of badly worded/constructed sentences. e.g., e.g. see the abstract. The manuscript needs a careful and professional proofread.
Response 2: The grammatical errors in the paper have been corrected.
Comment 3: Need to clearly define all expressions and abbreviations the first time they are used; e.g. what is “Oral and Media” in Fig. 1? What is BASS in section 2?
Response 3: Nomenclature is added before the first part. All expressions and abbreviations are clearly defined in the paper . “Oral and Media” and “BASS” are clearly defined in the second paragraph of section 1.
Comment 4: Ras in lower box of Fig. 1 should be RAs.
Response 4:The Ras in the lower box of Figure 1 has been changed to RAs, and the corresponding errors in the paper have also been revised.
Comment 5: Axes of graphs, e.g. Fig.3 should be appropriately labeled. Also, the demand profiles for the weekday and weekend in (a) and (b) seem a bit unusual. Shouldn’t these be the other way round, this is: (a) weekend and (b) weekday?
Response 5: Axes of graphs, e.g. Fig.3 have be labeled. The data of Fig. 3 is from a power company in China. Fig. 3 shows the typical per unit curve of daily load power in residential areas, and the reference value is 700kW.
Comment 6: I have not come across the term “charging pile”. In the literature this is usually referred to as charging post, point or station.
Response 6: The term "charging pile" in the paper has been changed to "charging post", and the corresponding errors have also been revised.
Comment 7: What does “Combining (2)” on page 9 means?
Response 7: The sentence on page 9 has been changed as “Using formula (2) and the ownership of private vehicles in 2019, the saturation value of private vehicle ownership in the residential area can be calculated as 2762.”
Comment 8: Please check conclusions 2, Line 2: I think “greater” is wrong.
Response 8: The word “greater” has been changed as “bigger”.
Comment 9: There are significant number of publications on the impact of EV charging on the grid. Authors are advised to enhance the literature review, to clearly demonstrate the added value and new knowledge that this paper presents. Just to give a few examples:
- Das R., et al, “Real-time multi-objective optimisation for electric vehicle charging management”, Elsevier journal of Cleaner Production, Vol. 292, 10 April 2021 126066, pp 1-19.
- Lacey G., et al, “Smart EV Charging Schedules: Supporting the Grid and Protecting Battery Life", IET Electrical Systems in Transportation, Vol. 7, Issue 1, March 2017, pp 84-91.
- Jiang T., et al, “Development of a Decentralized Smart Charge Controller for Electric Vehicles”, Elsevier International Journal of Electrical Power & Energy Systems, Volume 61, October 2014, pp. 355–370.
- Putrus G. A., et al, “Impacts of Electric Vehicles on Power Distribution Networks” IEEE Vehicle Power and Propulsion Conference, Michigan, Sept. 2009, pp 827-831.
Please note that these are only examples, and you may include what you believe is relevant. The important main point is to better discuss the state-of-the-art and to put the interpretation of the results into the right context.
Response 9: We revised the literature review, and highlighted the motivation and contribution of this paper. Also, relevant literature is added in the reference part.

Reviewer 2 Report
An orderly charging method for planning the charging piles and distribution net work of RAs is proposed in this paper. Comments to the authors:
1) This paper is organized as follows. Section II ....... change the section numbers to 2, 3, 4, etc. in the organization section.
2) Explanation about forecasting of charging load can be provided.
3) How the uncertainty of EVs has been handled in this work.
4) Please provide the test system details.
5) The motivation of the study should be better highlighted.
6) Some grammar mistakes exist in this paper.
7) The drawbacks of the existing methods must be highlighted clearly for justfying the necessity of the proposed work.
Author Response
Manuscript ID: wevj-1400497
Title: A Medium and Long Term Orderly Charging Load Planning Method for Electric Vehicles in Residential Areas
Journal: World Electric Vehicle Journal
Mr. Nopparat Komrapit, Assistant Editor, World Electric Vehicle Journal
First of all, we would like to thank you, the assistant editor and the reviewers for your insightful and helpful comments. We have carefully read the comments from you and the reviewers, and we have revised our paper according to the comments and suggestions. The changes in the manuscript are marked up using the “Track Changes”.
The authors.
Reviewer 2:
An orderly charging method for planning the charging piles and distribution network of RAs is proposed in this paper. Comments to the authors:
Comment 1: This paper is organized as follows. Section II ....... change the section numbers to 2, 3, 4, etc. in the organization section.
Response 1: Section II, III,IV… have been changed to Section 2, 3, 4, etc.
Comment 2: Explanation about forecasting of charging load can be provided.
Response 2: The forecasting steps of charging load is added in the paper as follows:
(1). Use the Bass model to forecast the medium and long-term ownership of electric vehicles in residential areas;
(2). According to the established mathematical model of electric vehicle driving characteristics, a large number of samples are randomly selected;
(3). Calculate the charging start SOC and charging time of each random sample;
(4). Count whether each random sample is in a charging state at each moment;
(5). According to the statistical results of step 4, obtain the daily charging probability curve of electric vehicles;
(6). Loop the above steps 2 to 5 many times to obtain the average curve of the daily charging probability of electric vehicles;
(7). Combine the mid- and long-term electric vehicle ownership obtained in step 1 with the electric vehicle charging probability curve obtained in step 6, to obtain the number of electric vehicles charged at each moment;
(8). The number of electric vehicles charged at each moment in step 7 is multiplied by the charging power to obtain the electric vehicle charging load in residential areas.
Comment 3: How the uncertainty of EVs has been handled in this work.
Response 3: The probability density distribution function is used to deal with the uncertainty of electric vehicles. There are uncertainties in the starting time of electric vehicle charging and the mileage of electric vehicles. Through the collected private vehicle travel data, the probability density function of the private vehicle travel characteristics is fitted. In addition, repeated sampling is performed according to the fitted probability density distribution function to obtain the travel pattern of electric vehicles.
Comment 4: Please provide the test system details.
Response 4: The parameters of the test system are added in Appendix A. Table A1 is the load parameters and node capacity of the IEEE-33 node system, and Table A2 is the line parameters of the IEEE-33 node system.
Comment 5: The motivation of the study should be better highlighted.
Response 5: The motivation has been highlighted in Section 1.
Comment 6: Some grammar mistakes exist in this paper.
Response 6: The grammatical errors in the paper have been corrected.
Comment 7: The drawbacks of the existing methods must be highlighted clearly for justfying the necessity of the proposed work.
Response 7: Three models of EV ownership forecast are in the following: discrete model, multi-agent model and diffusion model. The definition is added in the second paragraph of Section 1. The application fields of short-term and medium- and long-term charging load forecasting are added in the second paragraph of Section 1. The orderly charging optimization strategy of electric vehicles can be divided into direct optimization method and indirect optimization method. The definition is added in the third paragraph of Section 1.

Round 2
Reviewer 1 Report
The authors have improved the manuscript, but the following comments have not been adequately addressed:
Comment 1
I am not convinced with the authors’ response and do not agree with their claim that “At present, cost, mileage anxiety, and the availability of charging infrastructure are not the main factors affecting the ownership of electric vehicles in urban residential areas.”.
I think the response given and assumption made (EVs as durable goods such as TVs and refrigerators) is too simplistic. The authors should, at least, elaborate on this and briefly explain whether and how these three factors (cost, range anxiety, availability of charging infrastructure) influence the analysis, results presented, and conclusions made.
Comment 2
The authors have improved the use of English, but I still think that the manuscript needs a careful and professional proofread. See the first sentence in the abstract (Line 8) and also the 5th sentence (Line 13). The 5th sentence is too long (over 6 lines) and difficult to follow and understand.
Comment 8
What I meant is: “After the implementation of the orderly charging strategy, the peak load of the EV access node will be smaller (or lower but not greater or bigger) than the peak power of disorderly charging. Is this right?
Author Response
Manuscript ID: wevj-1400497
Title: A Medium and Long Term Orderly Charging Load Planning Method for Electric Vehicles in Residential Areas
Journal: World Electric Vehicle Journal
Mr. Nopparat Komrapit, Assistant Editor, World Electric Vehicle Journal
First of all, we would like to thank you, the assistant editor and the reviewers for your insightful and helpful comments. We have carefully read the comments from you and the reviewers, and we have revised our paper according to the comments and suggestions. The changes in the manuscript are marked up using the “Track Changes”.
The authors.
Reviewer 1:
The authors have improved the manuscript, but the following comments have not been adequately addressed:
Comments:
Comment 1: I am not convinced with the authors’ response and do not agree with their claim that “At present, cost, mileage anxiety, and the availability of charging infrastructure are not the main factors affecting the ownership of electric vehicles in urban residential areas.”.
I think the response given and assumption made (EVs as durable goods such as TVs and refrigerators) is too simplistic. The authors should, at least, elaborate on this and briefly explain whether and how these three factors (cost, range anxiety, availability of charging infrastructure) influence the analysis, results presented, and conclusions made.
Response 1: The internal oral spreading factors include the purchase cost of electric vehicles, mileage anxiety, the availability of charging facilities, and the visibility of electric vehicles. It is considered as the imitation coefficient q. The context is added from line 67 to 71 in section 1 highlighted in yellow shadow.
Comment 2: The authors have improved the use of English, but I still think that the manuscript needs a careful and professional proofread. See the first sentence in the abstract (Line 8) and also the 5th sentence (Line 13). The 5th sentence is too long (over 6 lines) and difficult to follow and understand.
Response 2: The grammatical errors and the long sentences in the paper have been corrected.
Comment 8: What I meant is: “After the implementation of the orderly charging strategy, the peak load of the EV access node will be smaller (or lower but not greater or bigger) than the peak power of disorderly charging. Is this right?
Response 8: Yes, it is right. From Figure 6 (e), (f), (g) and (h), it can be seen that the orderly charging peak power may be bigger than the peak power of disorderly charging. It is because the charging load power will be shifted after the implementation of the orderly charging strategy and the total charging load of each EV access node is required to meet the optimal charging load power transmitted from the upper layer to the lower layer. If the peak load power of each EV access node is over the node capacity, the number of EV access nodes will be increased. The context is added from line 369 to 373 and from line 434 to 438 highlighted in yellow shadow.
